# Effects of Region, Processing, and Their Interaction on the Elemental Profiles of Pu-Erh Tea

**DOI:** 10.3390/foods14162848

**Published:** 2025-08-17

**Authors:** Yan-Long Li, He-Yuan Jiang, Ming-Ming Chen, Xiao-Li Wang, Hong-Yan Liu, Hai-Dan Zou, Bo-Wen Zhang, Ya-Liang Xu, Li-Li Qian

**Affiliations:** 1Institute of Urban Agriculture, Chinese Academy of Agricultural Sciences, Chengdu 610213, China; a99008191@163.com (Y.-L.L.); jiangheyuan@caas.cn (H.-Y.J.); zellazou@foxmail.com (H.-D.Z.); abc906769950@163.com (B.-W.Z.); xuyaliang@caas.cn (Y.-L.X.); 2College of Food Science, Heilongjiang Bayi Agricultural University, Daqing 163319, China; chenmingming515@163.com; 3College of Food and Bioengineering, Xihua University, Chengdu 610039, China; 15228782533@139.com

**Keywords:** food traceability, elemental fingerprinting, chemometrics, ICP-MS

## Abstract

Elemental contents are effective fingerprints for Pu-erh tea’s geographical traceability, which is crucial for consumer protection and sustainable development. Region and processing methods are key factors influencing the tea’s elemental fingerprint. This study analyzed 28 elements in Pu-erh tea samples from three Yunnan production regions subjected to different processing stages in the year of 2023. The results show that significant regional differences were observed for 25 of the 28 elements. As, Li, Cu, Zn, and Cd contents vary significantly during processing. The contents of 27 elements (excluding Pb) are significantly influenced by the interaction between region and processing stage. Orthogonal partial least squares discriminant analysis (OPLS-DA) achieved good validation (Q^2^ = 0.946) and identified 18 key factors, while the original and cross-validation correct classification rates were 100% and 98.6%, respectively. Crucially, the robustness of the model was confirmed with 100% accuracy through an independent validation set from tea samples in the harvest year of 2024. This study confirms that the elemental contents of Pu-erh tea are mainly influenced by region rather than processing stage, and elemental analysis can trace the geographical origin of Pu-erh effectively, even when mixed with a differently processed tea.

## 1. Introduction

Pu-erh tea is a special variety of tea originating from Yunnan province and is renowned worldwide for its distinctive rich flavor, intense aged aroma, and abundant health benefits [1,2,3,4,5]. Pu-erh tea can be categorized into two main types: raw Pu-erh and ripe Pu-erh tea. Ripe Pu-erh tea is usually manufactured through pile fermentation of fresh tea leaves [6]. During fermentation, substances such as catechins, caffeine, and amino acids present in the tea leaves undergo significant changes [7], making the tea taste smoother and developing a unique aged aroma. In recent years, with the rising popularity and increasing market demand for high-value Pu-erh tea, its similar appearance among different grades often leads to disruption in the tea market rules of fair trade [8]. This prevalence of counterfeit products with fake labels and falsified origin information can severely damage the reputation of authentic tea brands [9]. This situation erodes consumer trust, making the establishment of effective traceability techniques crucial for quality assurance and consumer protection [10].

With the rapid development of instrument technology and advancements in software and chemometrics methods, new technologies are gradually being employed for tea origin traceability [11], including spectroscopic techniques [12], chromatographic techniques [13], mass spectrometry techniques [14], and sensing technologies [15]. Among these, inductively coupled plasma mass spectrometry (ICP-MS) plays a crucial role in food geographical origin traceability [16], and its effectiveness in discriminating the geographical origins of various tea types has been widely confirmed. For instance, Luo et al. measured the content of 47 mineral elements in green tea from different regions of Guizhou, achieving an accuracy rate of 89.70% in distinguishing different origins [17]. Deng et al. determined the contents of 18 mineral elements (Rb, Mg, K, etc.) in tea leaves from major green tea-producing areas and used a random forest model for discrimination analysis, achieving an accuracy rate of 97.90% [18]. In addition, Zhang et al. and Ren et al. investigated the core production area of Qimen black tea, with both of these studies achieving 100% accuracy in origin classification [19,20].

Although elemental traceability based on geographical origin is effective, the stability and accuracy of the models are constrained by other key factors, such as tea plant varieties and processing techniques [21]. First of all, the elemental contents might change among different tea varieties. For example, Zhao et al. measured the contents of 16 mineral elements in fresh leaves of different tea varieties (‘Anji white’, ‘Wuniuzao’, and ‘Longjing 43’). They found significant differences (*p* < 0.05) in the contents of Mg, Al, V, Cu, Cs, Ba, Pb, and U among these varieties [22]. Mckenzie et al. measured 14 mineral elements in assorted teas (Pu-erh tea and others), finding that the contents of Na, Mg, Zn, Ca, Fe, and S in Pu-erh tea differed relative to four other tea types, making it possible to distinguish various types of tea [23]. In addition to variety, the processing technique, especially fermentation, is a key factor influencing the elemental fingerprint of ripe Pu-erh tea. For instance, Zhu et al. found that pile fermentation leads to significant enrichment of the vast majority of elements in tea leaves [24]. This drastic change introduced by processing undoubtedly interferes with the effectiveness of element fingerprints, posing a core challenge in the traceability research of ripe Pu-erh tea. Despite these challenges, similar successful approaches have been developed for the geographical discrimination of other complex samples, like coffee [25] and wine [26], which validates the potential of multi-element profiling for origin authentication.

Although previous studies have investigated the effects of geographical origin or processing techniques on tea’s elemental fingerprints separately, few have systematically evaluated the interaction between these two factors. This represents a significant knowledge gap, as the changes induced by processing (especially pile fermentation of ripe Pu-erh tea) can potentially mask or alter the original geographical signatures, posing a major challenge for authentic traceability. It remains unclear which elemental markers are robust enough to maintain their geographical specificity throughout the entire production chain.

To address this issue, this study aims to systematically investigate the effects of geographical region, processing stage, and their interaction on the elemental profiles of Pu-erh tea. The goal is to identify key elemental biomarkers that are stable and specific to geographical origin despite the influence of processing, and subsequently to establish and externally validate a robust discriminant model for geographical traceability accurately. This research will provide an experimental basis and candidate markers for constructing more reliable and universally applicable traceability models for ripe Pu-erh tea.

## 2. Materials and Methods

### 2.1. Materials

The three main production areas within the limited geographical indication area for Pu-erh tea (Jinggu County and Bangdong Township in Lincang City and Ning’er County in Yunnan Pu’er City) were selected for study in 2023 [27]. In each of the production areas, two representative tea gardens were chosen for sample collection. To ensure regional representativeness and minimize the idiosyncratic influence of a single plantation, fresh leaves from the gardens within each specific region were combined and thoroughly mixed to create a single, homogenized batch for that region. The ‘bud’ leaves (consisting of a bud, along with two leaves) from large-leaf tea species were collected for each sample. Detailed geographical information, including longitude, latitude, and altitude data, was recorded and is listed in Table 1. After collection, about 250 g of tea leaf samples were immediately transferred to the laboratory on dry ice for determination. Other tea leaves were used for further processing.

To externally validate the robustness and predictive power of the established discriminant model, an independent set of validation samples was collected again in the second year during the same season from the same three representative tea plantations. In 2024, a total of 18 tea samples (both fresh tea leaves and sun-dried tea leaves) were collected in these three regions. The analytical methods for all samples were identical to those used for the first batch of samples.

### 2.2. Fermentation of Ripe Pu-Erh Tea

The three homogenized regional batches underwent an identical pile fermentation process independently in the autumn of 2023. The specific fermentation procedures were as follows: after fixation, rolling, and sun-drying, the tea leaves were placed in special tea leaf piles. During the pile fermentation process, the temperature and humidity were controlled and recorded (Appendix A). The piles were regularly turned, with the process involving a total of four turnings to ensure an even exposure of the tea leaves to air, so as to promote the fermentation and oxidation processes.

To investigate the effect of processing on elemental contents, samples were systematically collected throughout this procedure. Sampling was conducted at eight key stages: fresh leaves, kill-green, rolling, first turning, second turning, third turning, fourth turning, and the final ripe Pu-erh tea product. At each of these eight stages, three independent replicate samples were taken from the fermentation pile. This experimental design resulted in a total of 24 analytical samples per region (8 processing stages × 3 replicates), providing the basis for the subsequent statistical analyses.

The tea samples were collected and pretreated as follows: first, the leaves were cleaned by removing dust, and then all the tea samples were dried at 40 °C to a constant weight. The dried samples were then finely ground into a fine powder with a miller (IKA, TUBE-MILL 100, Staufen, Germany). This powder was passed through a 100-mesh sieve to ensure a uniform particle size. Finally, all powdered samples were sealed in sample bags and stored at 4 °C for further analysis.

### 2.3. Multi-Element Analysis

The sample digestion method was conducted in line with a previous report with a slight modification [28]. About 0.25 g of the sample was processed with 6 mL of concentrated HNO_3_ (guaranteed reagent, Sinopharm Chemical Reagent Co., Ltd., Shanghai, China) in a Teflon digestion vessel for 2 h. Subsequently, 2 mL of BV-III-grade H_2_O_2_ (Sinopharm Chemical Reagent Co., Ltd., Shanghai, China) was added to the vessel and allowed to react for 30 min. After the release of nitrogen oxides, the digestion vessel was placed into a microwave digestion instrument (CEM MARS Xpress, Charlotte, NC, USA) and gradually heated to 180 °C for 40 min. After cooling, the digested solution was transferred to a 50 mL volumetric flask and diluted to the mark with ultrapure water. The solution was then filtered through a 0.22 μm membrane filter before ICP-MS analysis.

ICP-MS (Agilent, 8900, Santa Clara, CA, USA) was employed to measure the concentrations of twenty-eight isotopes (^7^Li, ^39^K, ^43^Ca, ^45^Sc, ^56^Fe, ^59^Co, ^63^Cu, ^66^Zn, ^75^As, ^85^Rb, ^88^Sr, ^89^Y, ^111^Cd, ^133^Cs, ^137^Ba, ^139^La, ^146^Nd, ^147^Sm, ^153^Eu, ^157^Gd, ^159^Tb, ^163^Dy, ^165^Ho, ^166^Er, ^169^Tm, ^172^Yb, ^175^Lu, and ^206^Pb). The operational parameters were set as follows: radio frequency power was set at 1280 W, and the atomizing chamber temperature was maintained at 2 °C. The flow rates for the cooling gas, carrier gas, and auxiliary gas were 1.47 L min^−1^, 1 L min^−1^, and 1 L min^−1^, respectively. To ensure accuracy, the certified reference material (CRM) of tea flour [29] underwent the same digestion and determination procedures as the sample.

To correct for matrix effects and instrument drift, a mixed internal standard solution containing Sc, Y, Rh, In, Tb, Lu, and Bi was used at a concentration of 10 μg/L (Sigma-Aldrich, St. Louis, MO, USA). All sample analyses were performed in triplicate, and re-measurement was conducted if the relative standard deviation of the internal standard concentration exceeded 5%. The performance of the analytical method was validated by determining the limit of detection (LOD), limit of quantification (LOQ), and recovery rate. The LOD and LOQ were determined from the analysis of eleven reagent blanks. The LOD was defined as three times the standard deviation of the blank signals (LOD = 3 × SD blank), while the LOQ was defined as ten times the standard deviation (LOQ = 10 × SD blank). Recovery experiments were conducted by spiking tea samples with known amounts of standard solutions. The recovery rate was calculated as the percentage of the detected amount of the spike relative to the added amount. The detailed results for LOD, LOQ, and recovery are presented in Appendix A.

### 2.4. Statistical Analysis

Statistical analyses of the data, including one-way analysis of variance (one-way ANOVA), multiway analysis of variance (two-way ANOVA), principal component analysis (PCA), and linear discriminant analysis (LDA), were carried out with SPSS for Windows version 22.0 (SPSS Inc., Chicago, IL, USA).

One-way ANOVA was applied to the elements to test for significant differences among the geographical origins and the processing stages. Levene’s test was first conducted to assess the homogeneity of variances. When variances were equal (*p* > 0.05), Tukey’s honestly significant difference (HSD) test was used for post hoc multiple comparisons. When variances were not equal (*p* < 0.05), Tamhane’s T2 test was used. Two-way ANOVA was applied to quantify the contributions of geographical origin, processing stage, and their interactions to the total variance in element levels. A factor with a larger ratio of relative variance indicates greater influence relative to the other factors. This analysis is crucial for screening robust traceability markers, which should ideally show the highest variance contribution for region.

PCA is a data dimensionality reduction technique that projects data onto a new coordinate system through linear transformation, while preserving the main information [30]. LDA is a supervised learning method for feature extraction and classification, aiming to find the optimal projection direction by maximizing the between-class distance and minimizing the within-class distance. LDA can effectively distinguish tea from different producing areas with effective parameters.

## 3. Results

### 3.1. The Difference in Multi-Elements Among Different Regions

The mean values and standard deviations (SD) of the mineral element contents in tea samples from different origins are shown in Table 2. Except for Cu, As, and Pb, all measured elements showed significant differences among the three geographical origins. Among them, Li showed a significant difference (*p* < 0.05), while the remaining 24 elements were highly significant (*p* < 0.01). It is worth noting that Zn was significantly lower in Jinggu compared with other regions, while for Cs, the significance was driven by lower concentrations in Ning’er compared to the other two regions (Table 2). Each origin exhibited distinct elemental profiles. Tea from Jinggu was particularly rich in Li, Sc, Y, Nd, Sm, Eu, Gd, Tb, Dy, Ho, Er, Tm, Yb, and Lu. In contrast, the concentrations of K, Ca, Fe, Co, Rb, Sr, Cd, Cs, Ba, La, and Pb were highest in tea sourced from Bangdong.

### 3.2. The Difference in Multi-Elements Among Different Processing Techniques

The mean values and standard deviations (SDs) of the mineral element contents of tea in various processing stages are shown in Table 3. Notably, the difference in As content was statistically significant (*p* < 0.05). Additionally, the differences in four elements (Li, Cu, Zn, and Cd) were highly significant (*p* < 0.01). Each processing stage exhibited distinct elemental profiles. For instance, fresh tea leaves were particularly rich in Sr and Ba, while Cs concentrations peaked during the kill-green stage. Compared to fresh leaves, the content of most mineral elements at the final ripe Pu-erh tea stage increased. The increases for Fe, Co, and Pb were particularly notable, rising by approximately 58% (from 86 to 136 mg/kg), 64% (from 103.06 to 168.65 µg/kg), and 83% (from 116.50 to 212.73 µg/kg), respectively. However, not all elements increased during processing. For instance, Sr and Ba content slightly decreased by approximately 4% (from 11.8 to 11.3 mg/kg) and 15% (from 27 to 23 mg/kg), respectively, while Cs content increased initially and then declined at the final stage.

The detailed mineral element contents for each of the three regions at eight distinct processing stages are presented in Appendix A. A key finding from this detailed analysis is that the trends of elemental changes during processing were not consistent across the three regions, revealing a significant interaction effect between region and processing. For example, the content of Cs showed a dramatic early-stage peak in the Bangdong region, but this pattern was not observed in the other two regions. Conversely, the Ba content was highest in fresh leaves in Bangdong, whereas it exhibited the highest level at the kill-green stage in Jinggu. Furthermore, elements such as La exhibited a general enrichment trend in Bangdong, whereas their concentrations remained relatively stable in Jinggu. These results clearly demonstrate that the elemental variations attributable to geographical origin discrimination are more pronounced and stable than those introduced by the processing stages.

### 3.3. Effects of Region, Processing Method, and Their Interaction on Multi-Elements

The region and processing method were considered as fixed factors. The effects were partitioned into different sources: region (R), processing method (P), and the interaction between region and processing method (R × P). The results show that most elements were significantly influenced by all three factors (R, P, and R × P). As detailed in Appendix A, ‘region’ was a significant factor for all elements, while ‘processing’ was significant for all elements except for Tb. The interaction effect (R × P) was significant for all elements except for Pb.

### 3.4. Principal Component Analysis

An initial exploratory analysis was conducted on the entire dataset of 28 elements using PCA to visualize the natural clustering of samples. The results are presented in Appendix A. As a result, three principal components were derived, accounting for a cumulative contribution of 86.76%. PC1, which explained 62.26% of the variance, had high positive loadings for nearly all 28 elements. In contrast, PC2 (14.93%) and PC3 (9.57%) had high loadings on a subset of elements. For example, PC2 showed high loadings for Zn, while PC3 was strongly influenced by a subset of rare earth elements and Cd. This suggests that while an overall multi-element gradient separates the teas (PC1, possibly correlating with overall soil mineral richness), there are secondary patterns driven by specific elements.

Notably, samples from three regions of origin (Jinggu, Bangdong, and Ning’er) were accurately distinguished, reflecting their distribution across distinct spatial regions. This demonstrates that the selected mineral elemental fingerprint can effectively differentiate Pu-erh tea from various origins (Figure 1).

### 3.5. Orthogonal Partial Least Squares Discriminant Analysis (OPLS-DA) of Pu-Erh Tea

In this study, an OPLS-DA discriminant model was constructed based on 28 indicators (Li, K, Ca, Sc, Fe, Co, Cu, Zn, As, Rb, Sr, Y, Cd, Cs, Ba, La, Nd, Sm, Eu, Gd, Tb, Dy, Ho, Er, Tm, Yb, Lu, and Pb), in order to distinguish tea samples from three major production origins: Jinggu, Bangdong, and Ning’er. The model was established with two predictive and one orthogonal component. No outliers were detected or removed during the model construction. The cross-validation Q^2^ value reached 0.946 (Figure 2A), significantly exceeding the model’s validity threshold. After 200 permutation tests, the intercept of the Q^2^ regression line on the vertical axis was below zero, indicating that the model was not overfitted and the validation results are reliable. Based on the criterion of a variable importance in projection (VIP) value greater than 1, 18 key indicators were selected and showed good potential for geographical origin discrimination (Figure 2B): Rb (VIP = 1.181), Co (VIP = 1.161), Fe (VIP = 1.132), Ba (VIP = 1.091), Tb (VIP = 1.063), Lu (VIP = 1.061), Yb (VIP = 1.058), Eu (VIP = 1.057), Dy (VIP = 1.055), Ho (VIP = 1.049), Gd (VIP = 1.048), Tm (VIP = 1.041), Er (VIP = 1.040), Sm (VIP = 1.028), Ca (VIP = 1.026), Cs (VIP = 1.019), Sc (VIP = 1.003), and Sr (VIP = 1.002). It is noteworthy that a majority of these key indicators (10 out of 18) are rare earth elements (REEs), which is consistent with prior studies suggesting that REEs are effective geological tracers for authenticating the origin of agricultural products [22].

### 3.6. Discriminant Analysis of Pu-Erh Tea from Different Regions

We performed stepwise discriminant analysis based on the following elements: Li, K, Ca, Sc, Fe, Co, Cu, Zn, As, Rb, Sr, Y, Cd, Cs, Ba, La, Nd, Sm, Eu, Gd, Tb, Dy, Ho, Er, Tm, Yb, Lu, and Pb. Out of these, eleven indicators were selected to construct the discriminant model. Furthermore, we employed a cross-validation model to classify the Pu-erh tea samples from three regions, and the results are outlined in Table 4.

In this model, the original correct discrimination rate reached 100%. The cross-validation results show that the classification accuracy for samples from Bangdong (24/24) and Ning’er (24/24) was 100%, while the classification accuracy for the Jinggu region was 95.8% (23/24), resulting in an overall classification accuracy of 98.6% (71/72). This indicates that the model constructed from these eleven elements can accurately distinguish the tea samples.

The discrimination model formulas were as follows:Y_Jinggu_ = −2.34 × 10^−4^ Ca − 9.68 × 10^−4^ Fe − 7.82 × 10^−1^ Co + 8.416 × 10^−3^ Cu + 1.52 × 10^−2^ Zn − 2.79 × 10^−3^ Rb + 7.700 × 10^−3^ Sr + 1.21 Cd + 4.27 × 10^−1^ Cs + 3.58 Gd + 6.44 YbY_Bangdong_ = 4.75 × 10^−4^ Ca – 5 × 10^−6^ Fe + 1.29 × 10^−1^ Co − 2.34 × 10^−3^ Cu + 3.40 × 10^−3^ Zn − 2.77 × 10^−4^ Rb − 4.37 × 10^−3^ Sr + 1.67 Cd + 2.18 × 10^−1^ Cs + 1.36 Gd + 4.38 YbY_Ning’er_ = −5.07 × 10^−4^ Ca − 1.10 × 10^−3^ Fe − 1.75 Co + 1.24 × 10^−2^ Cu + 2.34 × 10^−2^ Zn − 2.56 × 10^−3^ Rb + 9.85 × 10^−3^ Sr + 2.48 Cd + 1.69 × 10^−1^ Cs + 3.17 Gd + 1.42 Yb

The obtained distribution of Pu-erh tea from various regions is presented in Figure 3. This figure clearly illustrates the distinct separation of the Pu-erh tea samples from the three regions, each occupying its own unique space. Furthermore, a notable spatial gap exists between the regions, which confirms that the selected region-related indicators are both precise and reliable.

The 18 independent samples collected in the second year were used as a validation set, and the previously established discriminant model was applied for prediction. The results show that for every validation sample, the production area corresponding to the highest discriminant score calculated by the model was consistent with its actual production area. The external validation results of the model indicate (Table 5) that all 18 validation samples were correctly classified.

## 4. Discussion

This study found that the mineral element levels in Pu-erh tea from Jinggu and Bangdong were relatively high. Specifically, tea from Jinggu showed the highest content of 14 elements (Li, Sc, Y, Nd, Sm, Eu, Gd, Tb, Dy, Ho, Er, Tm, Yb, and Lu). Specifically, the La contents in tea from Jinggu and Bangdong were 193.27 μg/kg and 225.10 μg/kg, respectively, while in the tea from the Ning’er, this value was only 61.21 μg/kg, which is consistent with the study by Liu et al. [31]. A comprehensive analysis of Pu-erh tea across three distinct production regions indicated that the top three mineral elements in terms of content are K (28,939.67 mg/kg), Ca (4512.00 mg/kg), and Fe (136.00 mg/kg), which aligns with the findings of Shi et al. [32]. The Fe contents ranged from 100 mg/kg to 167 mg/kg, also consistent with a previous report [33]. The differences in elemental contents in tea among different regions may be related to different geographical and climatic conditions, including temperature, rainfall, sunlight, and soil [34,35]. Chen et al. indicated that the variation trends of certain elements (Mn, Rb, Tb, and Dy) in Pu-erh tea are consistent with the characteristics of the soil in which they grow [28]. Furthermore, the plant itself actively participates in the process, as its root system exhibits selective absorption capabilities for different elements. For example, studies on other plant-derived products, such as carobs, have demonstrated that soil properties exert a significant influence on the final mineral composition, highlighting the critical role of the entire soil–plant system in determining the elemental fingerprint associated with a specific geographical origin [36].

In addition, this study also indicated that the levels of several minerals in Pu-erh tea changed during tea processing. Compared with fresh tea leaves, the mineral element contents in ripe Pu-erh tea were generally higher (except for Sr and Ba). Among these, the increases in Pb, Cd, and Co were particularly notable, with increases of 83%, 78%, and 64%, respectively. These elements presented significant enrichment [37]. Certain mineral elements undergo transformation and release, significantly increasing the mineral element content in ripe Pu-erh tea [38]. Furthermore, microorganisms, primarily *Aspergillus niger*, secrete cellulase and pectinase, which may efficiently break down the cell walls of the tea leaves to release fixed minerals. Simultaneously, the metabolic activities of these microorganisms produce a large amount of organic acids (such as citric acid and gallic acid), causing a sharp decrease in the pH of the fermentation pile and further strongly enhancing the solubilization of various mineral salts [39]. In a study by Zhang et al., the contents of eight elements in raw Pu-erh tea were lower than those in ripe Pu-erh tea (Ba, Li, and Sr) [40]. Chen et al. discovered that throughout the fermentation of tea, the levels of Fe, Mn, Zn, Pb, Cd, Cr, and As increased with the increase in the number of turnings [41].

The final traceability model was built upon 11 key elemental markers, whose reliability was supported by multiple, complementary analytical approaches. Firstly, the two-way ANOVA results confirm the stability of these markers as geographical indicators, as their variance was predominantly driven by the ‘region’ factor rather than processing. Secondly, the OPLS-DA model highlighted their importance, with a clear majority (8 out of 11) of these markers featuring among the top predictors with high VIP scores. Finally, the stepwise discriminant analysis independently affirmed their predictive performance. It selected this specific combination of 11 elements—including some with lower VIP scores but high synergistic value—as the optimal set for classification, achieving 100% accuracy on the validation set. This convergence of evidence from three different analytical perspectives—stability (ANOVA), importance (OPLS-DA), and optimal predictive combination (SDA)—strongly supports that the selected markers are robust and reliable biomarkers for the geographical traceability of Pu-erh tea.

A key finding of this study is the significant interaction between geographical origin and processing. The results from the one-way ANOVA, which analyzed all samples from the three regions together, indicate that only five elements were significantly affected by the processing stage alone. However, the two-way ANOVA indicated that region, processing stage, and their interaction had a significant impact on most elements. Specifically, region significantly affected all elemental contents, processing stage influenced 27 elemental contents (excluding Tb), and their interaction significantly affected 27 elements (excluding Pb). The apparent discrepancy between these two results is explainable. In the one-way analysis where all regions were combined, the very large variance introduced by the ‘region’ factor masked the more subtle effects of ‘processing’. In contrast, the two-way ANOVA mathematically partitions this total variance, separating the dominant effect of region from the effect of processing. This makes the two-way model more sensitive, thus revealing the significant impact of the processing stage on a much larger number of elements. Given that the ‘region’ factor demonstrated such a dominant influence on the elemental profiles, multivariate pattern recognition techniques were subsequently applied to build a classification model. Principal component analysis revealed that the tea samples from three production areas could be distinctly differentiated, and the high classification accuracy of the OPLS-DA (Q^2^ = 0.946) and SDA (98.6%) models demonstrates the effectiveness of multi-element coupling for geographical discrimination.

A key advantage of this study is the successful external validation of the model using an independent set of samples from a different year, which were evenly distributed among the three regions (six samples per region). The high external validation accuracy of 100% strongly demonstrates the excellent temporal stability of the key elemental markers identified in this research. This confirms that the established traceability model is robust and reliable, rather than an incidental finding applicable only to a specific batch of samples.

## 5. Conclusions

This study conducted a comprehensive analysis of the content of 28 elements in Pu-erh tea samples from different regions and processing stages. Through multifactor analysis of variance, samples significantly affected by region were selected in order to trace the origin of the tea. The selected elements underwent PCA and stepwise LDA and OPLS-DA. The OPLS-DA model reverified the effective differentiation power of these elements for geographical traceability. Eleven elements were selected by the stepwise LDA, and accuracy rates of 100% in the original discriminant and 98.6% in the cross-validation were achieved. Furthermore, 18 samples from the next year were analyzed and validated, and 100% of the samples were correctly classified according to their region. These results indicate that the process of identifying origin based on elemental content is both accurate and effective. This study has not only established a discriminant model with high accuracy but, more importantly, has proven the model’s robustness and reliability through external validation with independent samples from a different year, thus providing crucial data support for its potential practical application.

## Figures and Tables

**Figure 1 foods-14-02848-f001:**
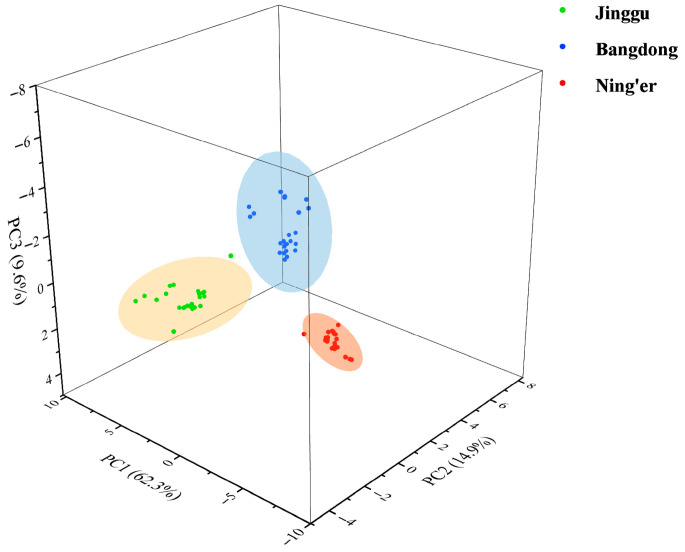
PCA scores of Pu-erh tea of different regions.

**Figure 2 foods-14-02848-f002:**
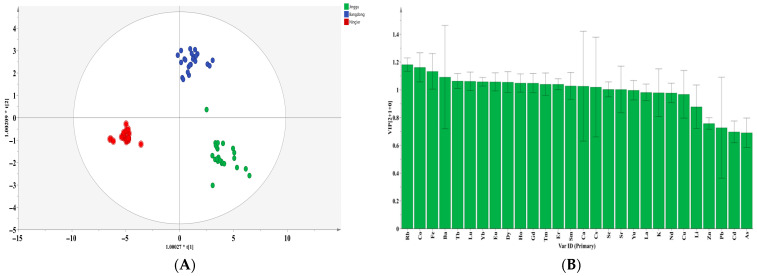
Scoring graph (**A**) and variable importance projection analysis (VIP) of the OPLS-DA model (**B**) based on mineral elements in different regions of Yunnan Province.

**Figure 3 foods-14-02848-f003:**
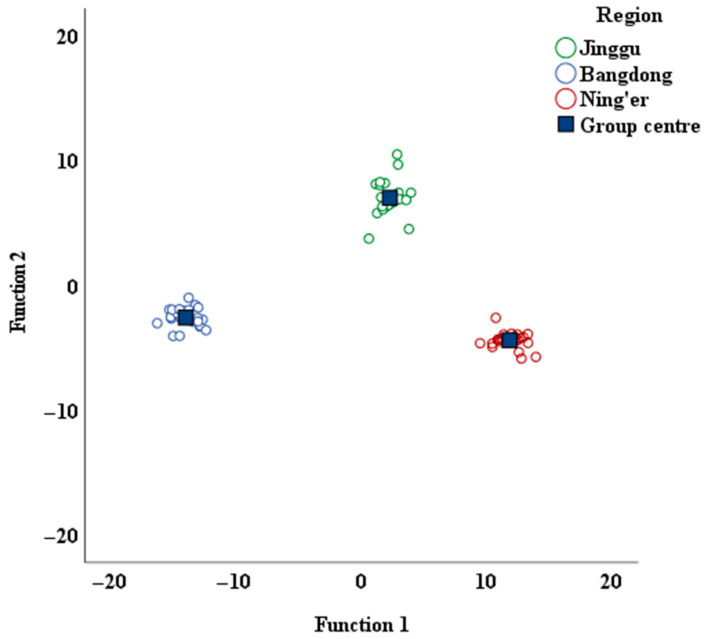
Discriminant function score map of Pu-erh tea from different regions.

**Table 1 foods-14-02848-t001:** The sample numbers, location, and tea geographical origins.

Region	Number of Samples	N Latitude (deg)	E Longitude (deg)	Altitude (m)
Jinggu	24	22.29–23.52	100.02–101.07	1842–1901
Bangdong	24	23.29–24.16	99.49–100.26	1633–1739
Ning’er	24	22.42–23.15	100.22–101.03	1330–1614

**Table 2 foods-14-02848-t002:** Mineral contents in Pu-erh tea from different regions.

Element	Jinggu	Bangdong	Ning’er
Li (ug/kg) *	78.14 ± 4.90 ^a^	74.47 ± 2.89 ^a^	67.21 ± 2.20 ^b^
K (mg/kg) **	28,476 ± 1103 ^b^	33,104 ± 475 ^a^	25,239 ± 469 ^c^
Ca (mg/kg) **	4572 ± 216 ^b^	5501 ± 134 ^a^	3463 ± 24 ^c^
Sc (ug/kg) **	48.600 ± 3.349 ^a^	33.501 ± 4.357 ^b^	16.889 ± 1.675 ^c^
Fe (mg/kg) **	141 ± 9 ^b^	167 ± 8 ^a^	100 ± 1 ^c^
Co (ug/kg) **	147.41 ± 6.80 ^b^	286.94 ± 10.21 ^a^	71.59 ± 0.57 ^c^
Cu (mg/kg)	18.1 ± 7.3	17.7 ± 0.2	17.7 ± 0.3
Zn (mg/kg) **	36 ± 1 ^b^	39 ± 1 ^a^	40 ± 0 ^a^
As (ug/kg)	64.66 ± 11.62	63.70 ± 9.64	55.04 ± 7.48
Rb (mg/kg) **	109.5 ± 5.0 ^b^	161.3 ± 2.3 ^a^	46.7 ± 0.4 ^c^
Sr (mg/kg) **	13.5 ± 0.6 ^b^	14.7 ± 0.3 ^a^	5.7 ± 0.1 ^c^
Y (ug/kg) **	268.7 ± 14.2 ^a^	219.4 ± 4.2 ^b^	71.7 ± 1.9 ^c^
Cd (ug/kg) **	50.865 ± 2.508 ^b^	56.816 ± 1.500 ^a^	31.560 ± 1.061 ^c^
Cs (ug/kg) **	261.61 ± 12.76 ^ab^	278.54 ± 4.80 ^a^	184.52 ± 2.46 ^c^
Ba (mg/kg) **	18 ± 1 ^b^	38 ± 1 ^a^	13 ± 0 ^c^
La (ug/kg) **	193.27 ± 8.04 ^b^	225.10 ± 5.15 ^a^	61.21 ± 2.52 ^c^
Nd (ug/kg) **	188.48 ± 4.54 ^a^	150.63 ± 4.56 ^b^	51.25 ± 0.43 ^c^
Sm (ug/kg) **	42.641 ± 3.859 ^a^	26.959 ± 1.841 ^b^	11.534 ± 1.406 ^c^
Eu (ug/kg) **	10.732 ± 0.630 ^a^	7.389 ± 1.047 ^b^	2.478 ± 0.298 ^c^
Gd (ug/kg) **	44.775 ± 2.725 ^a^	31.805 ± 1.304 ^b^	12.718 ± 0.012 ^c^
Tb (ug/kg) **	6.569 ± 0.520 ^a^	4.304 ± 0.199 ^b^	1.778 ± 0.115 ^c^
Dy (ug/kg) **	40.342 ± 3.705 ^a^	26.700 ± 0.469 ^b^	10.206 ± 0.496 ^c^
Ho (ug/kg) **	8.476 ± 0.202 ^a^	5.885 ± 0.289 ^b^	2.130 ± 0.190 ^c^
Er (ug/kg) **	25.253 ± 1.131 ^a^	17.345 ± 1.151 ^b^	5.595 ± 0.297 ^c^
Tm (ug/kg) **	3.533 ± 0.081 ^a^	2.570 ± 0.177 ^b^	0.698 ± 0.042 ^c^
Yb (ug/kg) **	24.558 ± 0.599 ^a^	16.130 ± 0.468 ^b^	4.629 ± 0.410 ^c^
Lu (ug/kg) **	3.636 ± 0.221 ^a^	2.383 ± 0.191 ^b^	0.661 ± 0.074 ^c^
Pb (ug/kg)	109.00 ± 27.71	152.04 ± 2.31	88.44 ± 6.09

Data are shown as the mean ± standard deviation. ^a–c^ in the same row indicate that there are significant differences among regions at *p* < 0.05 level, as determined by one-way ANOVA followed by Duncan’s multiple range test. * means significant difference (*p* < 0.05); ** means highly significant difference (*p* < 0.01).

**Table 3 foods-14-02848-t003:** Mineral contents in Pu-erh tea using different processing techniques.

Element	Fresh Leaves	Kill-Green	Rolling	First Turning	Second Turning	Third Turning	Fourth Turning	Ripe Pu-Erh Tea
Li (ug/kg) **	54.42 ± 33.59 ^abcd^	49.84 ± 12.34 ^bcd^	50.45 ± 11.60 ^bcd^	62.14 ± 3.75 ^bcd^	62.81 ± 8.41 ^abcd^	63.33 ± 7.51 ^abcd^	67.25 ± 14.07 ^abcd^	73.28 ± 5.70 ^a^
K (mg/kg)	23,943 ± 1910	23,709 ± 1989	23,625 ± 2402	24,595 ± 2393	26,550 ± 3966	26,097 ± 3387	27,072 ± 5331	28,940 ± 3483
Ca (mg/kg)	4320 ± 76	3780 ± 31	3750 ± 25	3540 ± 56	3990 ± 76	3820 ± 64	4020 ± 69	4510 ± 89
Sc (ug/kg)	26.580 ± 15.1120	28.923 ± 14.557	29.018 ± 17.925	29.036 ± 14.600	32.742 ± 14.988	31.183 ± 12.809	33.686 ± 14.302	32.997 ± 14.034
Fe (mg/kg)	86 ± 31	90 ± 21	90 ± 15	110 ± 22	119 ± 32	118 ± 30	135 ± 49	136 ± 30
Co (ug/kg)	103.06 ± 63.53	97.66 ± 34.52	92.66 ± 28.98	150.11 ± 86.14	160.39 ± 96.20	158.24 ± 91.07	169.79 ± 107.18	168.651 ± 94.799
Cu (mg/kg) **	14.6 ± 2.7 ^abcd^	12.7 ± 4.0 ^abcd^	13.7 ± 3.6 ^abcd^	16.1 ± 1.1 ^abcd^	16.7 ± 0.4 ^bcd^	16.6 ± 0.5 ^bcd^	16.9 ± 0.9 ^abcd^	18.0 ± 0.6 ^a^
Zn (mg/kg) **	32 ± 5 ^abcd^	32 ± 4 ^bcd^	31 ± 3 ^bcd^	34 ± 2 ^bcd^	36 ± 2 ^abcd^	36 ± 1 ^abcd^	37 ± 4 ^abcd^	38 ± 2 ^a^
As (ug/kg) *	41.29 ± 18.50 ^ab^	40.30 ± 11.420 ^b^	42.66 ± 9.97 ^ab^	49.56 ± 7.11 ^ab^	50.56 ± 5.51 ^ab^	51.60 ± 4.45 ^ab^	56.36 ± 8.97 ^ab^	61.14 ± 9.59 ^a^
Rb (mg/kg)	83.1 ± 39.0	89.0 ± 38.9	87.7 ± 37.9	93.8 ± 41.8	99.9 ± 49.4	97.8 ± 46.8	103.3 ± 53.8	105.8 ± 49.7
Sr (mg/kg)	11.8 ± 4.6	10.2 ± 5.5	9.7 ± 4.8	8.8 ± 3.3	10.0 ± 3.8	9.6 ± 3.7	10.5 ± 3.9	11.3 ± 4.2
Y (ug/kg)	147.9 ± 94.5	138.2 ± 98.2	145.9 ± 118.0	162.4 ± 82.2	175.5 ± 92.4	169.1 ± 81.3	179.5 ± 78.4	186.6 ± 89.0
Cd (ug/kg) **	26.031 ± 5.396 ^c^	31.505 ± 13.585 ^bc^	31.886 ± 12.544 ^bc^	40.335 ± 13.442 ^ab^	43.472 ± 11.799 ^ab^	40.257 ± 11.421 ^ab^	45.201 ± 13.223 ^a^	46.413 ± 11.539 ^a^
Cs (ug/kg)	193.98 ± 96.27	311.04 ± 149.49	299.80 ± 133.8	221.97 ± 45.54	222.04 ± 46.91	225.82 ± 43.37	232.78 ± 58.59	241.55 ± 43.95
Ba (mg/kg)	27 ± 12	18 ± 5	17 ± 3	19 ± 9	21 ± 11	21 ± 10	23 ± 11	23 ± 11
La (ug/kg)	154.22 ± 111.95	104.92 ± 51.81	97.56 ± 56.26	141.38 ± 71.25	153.96 ± 76.80	142.38 ± 71.54	159.46 ± 72.17	159.86 ± 75.42
Nd (ug/kg)	122.10 ± 87.20	101.67 ± 60.81	97.75 ± 74.67	117.65 ± 60.55	131.93 ± 65.19	118.23 ± 58.62	127.93 ± 56.50	130.12 ± 61.46
Sm (ug/kg)	25.711 ± 18.747	22.220 ± 13.236	21.477 ± 18.506	24.728 ± 13.645	27.187 ± 14.764	25.028 ± 13.470	27.323 ± 12.893	27.045 ± 13.656
Eu (ug/kg)	6.050 ± 3.614	5.349 ± 3.982	6.021 ± 5.101	6.000 ± 3.294	6.639 ± 3.437	6.180 ± 3.214	6.473 ± 3.224	6.866 ± 3.650
Gd (ug/kg)	25.415 ± 17.203	22.274 ± 14.719	23.759 ± 20.060	25.405 ± 13.441	27.278 ± 13.656	26.533 ± 13.038	27.655 ± 12.897	29.766 ± 14.046
Tb (ug/kg)	3.725 ± 2.550	3.591 ± 2.678	3.704 ± 3.223	3.865 ± 2.195	4.012 ± 2.109	4.080 ± 2.072	4.199 ± 1.904	4.217 ± 2.095
Dy (ug/kg)	21.968 ± 14.721	21.726 ± 16.686	23.135 ± 20.913	22.282 ± 12.896	24.917 ± 13.871	23.692 ± 12.720	25.790 ± 13.219	25.749 ± 13.204
Ho (ug/kg)	4.556 ± 3.080	4.512 ± 3.506	4.628 ± 4.093	4.766 ± 2.606	5.157 ± 2.990	4.941 ± 2.722	5.216 ± 2.653	5.497 ± 2.770
Er (ug/kg)	12.741 ± 8.602	13.037 ± 10.608	12.876 ± 11.813	14.137 ± 8.435	14.446 ± 8.151	14.579 ± 8.367	15.755 ± 8.427	16.065 ± 8.605
Tm (ug/kg)	1.931 ± 1.298	1.938 ± 1.706	1.962 ± 1.932	2.182 ± 1.323	2.115 ± 1.296	2.126 ± 1.254	2.343 ± 1.261	2.267 ± 1.252
Yb (ug/kg)	11.562 ± 7.479	12.745 ± 11.362	12.951 ± 12.77	13.914 ± 9.336	14.414 ± 9.229	14.025 ± 9.153	14.649 ± 8.755	15.105 ± 8.674
Lu (ug/kg)	1.762 ± 1.251	2.096 ± 1.948	1.891 ± 1.907	2.050 ± 1.362	2.040 ± 1.43	2.087 ± 1.275	2.210 ± 1.342	2.227 ± 1.303
Pb (ug/kg)	116.495 ± 31.506	128.709 ± 31.553	147.622 ± 25.115	162.377 ± 34.825	163.135 ± 32.215	163.067 ± 25.378	202.184 ± 16.909	212.732 ± 60.063

Data are shown as the mean ± standard deviation. ^a–d^ in the same row indicate that there are significant differences among processing stages at *p* < 0.05 level, as determined by one-way ANOVA followed by Duncan’s multiple range test. * means significant difference (*p* < 0.05); ** means highly significant difference (*p* < 0.01).

**Table 4 foods-14-02848-t004:** Classification results of Pu-erh tea from different regions.

		Region	Jinggu	Bangdong	Ning’er	Total
Original	count	Jinggu	24	0	0	24
		Bangdong	0	24	0	24
		Ning’er	0	0	24	24
	%		100	100	100	100
Cross-validation	count	Jinggu	23	0	1	24
		Bangdong	0	24	0	24
		Ning’er	0	0	24	24
	%		95.8	100	100	98.6

**Table 5 foods-14-02848-t005:** The external validation results based on the indicators of new Pu-erh tea samples.

Sample ID	Actual Group	Predicted Group	Y_Jinggu_ Score	Y_Bangdong_ Score	Y_Ning’er_ Score
Jinggu	Jinggu	Jinggu	**−262.152**	−408.835	−419.62
			**−267.437**	−414.939	−425.922
			**−268.049**	−414.898	−425.862
			**−266.368**	−411.391	−421.944
			**−268.026**	−414.881	−425.617
			**−264.444**	−413.433	−424.286
Bangdong	Bangdong	Bangdong	−409.529	**−313.67**	−356.126
			−409.006	**−312.333**	−354.276
			−411.512	**−314.777**	−357.514
			−410.97	**−314.24**	−356.843
			−414.776	**−316.326**	−358.374
			−414.285	**−316.59**	−359.186
Ning’er	Ning’er	Ning’er	−420.246	−357.652	**−323.08**
			−426.697	−361.328	**−325.295**
			−421.465	−356.671	**−321.491**
			−425.105	−364.505	**−331.439**
			−427.043	−362.274	**−326.632**
			−428.163	−364.421	**−328.718**

## Data Availability

The original contributions presented in this study are included in the article/Appendix A. Further inquiries can be directed to the corresponding authors.

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
