# Peer review of "Effects of Region, Processing, and Their Interaction on the Elemental Profiles of Pu-Erh Tea"

_foods, 2025, doi:10.3390/foods14162848_

Round 1
Reviewer 1 Report
Comments and Suggestions for Authors
Review on manuscript: foods-3775875
Effects of region, processing and their interaction on the multi-elements in Pu-erh tea
by Yan-Long Li, He-Yuan Jiang, Ming-Ming Chen, Xiao-Li Wang, Hong-Yan Liu, Hai-Dan Zou, Bo-Wen Zhang, Ya-Liang Xu, Li-Li Qian
submitted to Foods
In the manuscript submitted for evaluation, the authors studied the effects of region, processing and their interaction on the multi-elements in Pu-erh tea.
Overall, the manuscript is prepared correctly and requires only a few additions..
Detailed recommendation:
line 19 – abbreviations should be explained when first used,
lines 71-78 – the introduction should end with a clearly formulated research objective, not a summary of the methods used,
lines 82-84 – information on the number of samples and years of collection is unclear,
line 94 – what conditions were the samples stored in after harvest?
line 109 – what conditions were the samples stored before analysis?
line 118 – should be: USA,
lines 127-128 – how were recovery, LOD and LOQ determined?
lines 162 and 179 – different letters?
Author Response
Dear Editor,
Thank you for giving us a chance to improve the quality of our review paper, those comments are all valuable and very helpful for revising and improving our paper. The whole manuscript was revised according to the reviewers’ and editor’s suggestions. Revised portions are marked in red on the paper.
Thank you very much for considering our manuscript. I'm looking forward to hearing from you soon.
Best regards,
Hongyan Liu Ph.D.
Research Center for Plants and Human Health, Institute of Urban Agriculture, Chinese Academy of Agricultural Sciences, Chengdu National Agricultural Science & Technology Center, Chengdu 610213, China.
E-mail: liuhongyan01@caas.cn
Reviewer: 1
Comments to the Author
In the manuscript submitted for evaluation, the authors studied the effects of region, processing and their interaction on the multi-elements in Pu-erh tea.
Overall, the manuscript is prepared correctly and requires only a few additions.
- line 19 – abbreviations should be explained when first used.
Response: Thank you for your suggestion. We have given the full name of "OPLS-DA" when it first appeared in this article. Please see line 20.
- lines 71-78 – the introduction should end with a clearly formulated research objective, not a summary of the methods used.
Response: Thank you for your suggestion. The last part of the introduction has been modified as required. Please see line 84-91.
- lines 82-84 – information on the number of samples and years of collection is unclear.
Response: Thank you for your question. Parts 2.1 and 2.2 have been modified and more sample information has been added, The year information is in lines 96, The weight information of the original sample is shown in lines 103, and the number of experimental samples is shown in lines 122-128.
- line 94 – what conditions were the samples stored in after harvest?
Response: Thank you for your question. Details have been added. Please see line 103-106.
- line 109 – what conditions were the samples stored before analysis?
Response: Thank you for your question. Details have been added. Please see line 133-134.
- line 118 – should be: USA.
Response: Thank you for your suggestion. I have checked other contents and made some changes. Please see line 141 and 146.
- lines 127-128 – how were recovery, LOD and LOQ determined?
Response: Thank you for your question. The measurement method for these parameters has been supplemented. Please see line 158-167.
- lines 162 and 179 – different letters?
Response: Thank you for your question. The range of letters used for significance marking differs between Table 2 (a, b, c) and Table 3 (a, b, c, d) because the number of groups being compared in the one-way ANOVA is different. Table 2 compares 3 geographical regions, thus requiring a maximum of three letters to denote significant differences. In contrast, Table 3 compares 8 different processing stages, which may require four or more letters. We have double-checked our statistical analysis and confirm that the use of these letters is appropriate and correct in both tables.
Reviewer 2 Report
Comments and Suggestions for Authors
Thank you for the opportunity to participate in the revision of the article entitled “Effects of region, processing and their interaction on the multi-elements in Pu-erh tea”.
This study aimed to analyze the mineral content of Pu-erh tea samples from three production regions in Yunnan at different processing stages. In addition, discriminant analysis methods were employed in relation to the different geographical origin. The ultimate goal was to determine more robust geographic origin markers that are less influenced by processing, to provide a traceability model for ripe Pu-erh tea.
I think the authors set up the work well. The subject matter is interesting and explained clearly. In the introduction, they explain the origins of Pu-erh tea, and the issues related to potential fraud. This prompted the development of a traceability model for this product. The results of the study are also presented clearly, partly through the use of tables and graphs.
In conclusion, I would rate this “minor revisions”.
The following comments have been provided:
2. Materials and Methods
2.3. Multi-element analysis
Lines 114-117: “After the release of nitrogen oxides, the digestion vessels were placed into a microwave digestion instrument (CEM MARS Xpress, Charlotte, NC, America) and gradually heated to 180 °C for 40 min”. Were the samples diluted? If so, by how much? Did you filter them? I recommend including this information in the text.
Lines 125-127: “All sample analysis were performed in triplicate, a re-measurement was conducted if the relative standard deviation of the internal standard concentration exceeded 5%..”. Remove the double full stop at the end of the sentence.
4. Discussion
Lines 248-251: “The study found that the mineral element levels in Pu-erh tea from the Jinggu and Bangdong were relatively high, the results showed that the tea from Jinggu region had the highest content of 14 elements (Li, Sc, Y, Nd, Sm, Eu, Gd, Tb, Dy, Ho, Er, Tm, Yb, and Lu)”. I would suggest splitting this sentence into two.
Lines 257-262: “The differences of elemental contents in tea among different regions might be related to the different geographical and climatic conditions, including temperature, rainfall, sunlight and soil [28-29]. Chen et al. indicated that the variation trends of certain elements (Mn, Rb, Tb, and Dy) in Pu-erh tea were consistent with the characteristics of the soil in which they grow [23]”. That is correct; the mineral profile is influenced by all these variables. However, one must also consider the soil's chemical and physical characteristics, such as pH, which affect the availability of minerals. Consequently, one must also consider the plant's ability to absorb these elements from the soil. In this regard, I recommend citing the following articles:
https://doi.org/10.1016/j.jfca.2025.107628
Author Response
Dear Editor,
Thank you for giving us a chance to improve the quality of our review paper, those comments are all valuable and very helpful for revising and improving our paper. The whole manuscript was revised according to the reviewers’ and editor’s suggestions. Revised portions are marked in red on the paper.
Thank you very much for considering our manuscript. I'm looking forward to hearing from you soon.
Best regards,
Hongyan Liu Ph.D.
Research Center for Plants and Human Health, Institute of Urban Agriculture, Chinese Academy of Agricultural Sciences, Chengdu National Agricultural Science & Technology Center, Chengdu 610213, China.
E-mail: liuhongyan01@caas.cn
Reviewer: 2
This study aimed to analyze the mineral content of Pu-erh tea samples from three production regions in Yunnan at different processing stages. In addition, discriminant analysis methods were employed in relation to the different geographical origin. The ultimate goal was to determine more robust geographic origin markers that are less influenced by processing, to provide a traceability model for ripe Pu-erh tea.
I think the authors set up the work well. The subject matter is interesting and explained clearly. In the introduction, they explain the origins of Pu-erh tea, and the issues related to potential fraud. This prompted the development of a traceability model for this product. The results of the study are also presented clearly, partly through the use of tables and graphs.
In conclusion, I would rate this “minor revisions”
- “After the release of nitrogen oxides, the digestion vessels were placed into a microwave digestion instrument (CEM MARS Xpress, Charlotte, NC, America) and gradually heated to 180 °C for 40 min”. Were the samples diluted? If so, by how much? Did you filter them? I recommend including this information in the tex
Response: Thank you for your question, the mentioned information above has been added in section 2.3. Please see line 142-145.
- All sample analysis were performed in triplicate, a re-measurement was conducted if the relative standard deviation of the internal standard concentration exceeded 5%..”. Remove the double full stop at the end of the sentence
Response: Thank you for your suggestion. It has been modified. Please see line 158.
- The study found that the mineral element levels in Pu-erh tea from the Jinggu and Bangdong were relatively high, the results showed that the tea from Jinggu region had the highest content of 14 elements (Li, Sc, Y, Nd, Sm, Eu, Gd, Tb, Dy, Ho, Er, Tm, Yb, and Lu)”. I would suggest splitting this sentence into two.
Response: Thank you for your suggestion. It has been modified. Please see line 319-321.
- Lines 257-262: “The differences of elemental contents in tea among different regions might be related to the different geographical and climatic conditions, including temperature, rainfall, sunlight and soil [28-29]. Chen et al. indicated that the variation trends of certain elements (Mn, Rb, Tb, and Dy) in Pu-erh tea were consistent with the characteristics of the soil in which they grow [23]”. That is correct; the mineral profile is influenced by all these variables. However, one must also consider the soil's chemical and physical characteristics, such as pH, which affect the availability of minerals. Consequently, one must also consider the plant's ability to absorb these elements from the soil. In this regard, I recommend citing the following articles:https://doi.org/10.1016/j.jfca.2025.107628
Response: Thank you for your advice and the highly relevant literature provided. The discussion section has been revised. Please see line 333-338.
Reviewer 3 Report
Comments and Suggestions for Authors
The manuscript investigates the influence of geographic origin, fermentation processing, and their interaction on the multi-element composition of Pu-erh tea using ICP-MS, ANOVA, and multivariate statistical models to establish a reliable traceability approach validated across different harvest years. While the manuscript is generally interesting and holds sufficient interest for Foods, it has several limitations. I recommend a major revision of the text.
The title is somewhat long and can be more specific. The title uses the phrase “multi-elements,” which is grammatically odd. Consider rephrasing to “multi-element composition” or “multi-element profiles” for clarity and correctness or change title for example: “Effects of Region and Processing on Pu-erh Tea Elemental Profiles”.
Abstract:
Line 16: The wording “then processed with different processing methods” is repetitive and unclear.
Line 16-17: The sentence “significant differences were found in the content of 25 elements among different regions” is cumbersome. Consider: “significant regional differences were observed for 25 of the 28 elements.”
Line 19: . For clarity, define abbreviation OPLS-DA at first use.
Abstract vs. conclusion consistency (Lines 22-23 vs. Lines 305-307): The abstract mentions “100% accuracy through an independent validation set from different harvest years” (plural “years”), whereas the conclusion says “independent samples from a different year” (singular). This is slightly contradictory. If only one year of external samples was used (as it appears, 18 samples from the next year), change the abstract to “from a different harvest year” to match the actual procedure. Consistency here will avoid confusing readers about how many years of validation data were used.
The keywords include terms already present in the title: Pu-erh tea
Introduction
Line 30: Add “of tea” after “special variety” for clarity. Also, “found in Yunnan” could be “originating from Yunnan” to emphasize geographical origin.
Line 36: The participle “developing” does not agree with the subject. It should be “…making the tea taste smoother and develop a unique aged aroma.”
Line 36-41: References are missing.
Line 59-62: The sentence is unclear; it should be rephrased and possibly split into two separate sentences.
Line 56-70:To strengthen the rationale of using multi-element profiles for origin authentication and to place the current study within a broader context of similar approaches applied to other plant-derived products, it is recommended to consider citing additional literature where ICP-MS and elemental fingerprinting were successfully used for the geographical discrimination of beverages (https://doi.org/10.3390/foods14020275).
Lines 30-79: The manuscript would benefit from an explicit statement of the knowledge gap and objective. Currently, many references are cited about tea origin authentication, but it’s not clearly stated what has not been done. Improve novelty.
Material amd methods
Line 97: Sampling design clarity. It’s stated that 24 samples were collected per region (Table 1), and that leaves from each region were fermented separately. It’s unclear how those 24 samples were used in the fermentation experiment. Did each region’s 24 samples get combined into one pile, or were there multiple piles/replicates? Please clarify the experimental design: for example, if there were replicates of the fermentation per region or if those 24 samples correspond to, say, triplicate samples at each of 8 processing stages. This is important for understanding the statistical independence of the data.
Line 106: Awkward phrasing: “Placed in a dryer at 40℃ until constant weight” should be rephrased for clarity. Typically one says “dried at 40 °C to constant weight.”
Line 109: It should be “uniform particle size.”
Line 127: Internal standard details. You mention that an internal standard was used. However, the specific internal standard element(s) are not identified anywhere. Please specify which internal standard(s) you used for ICP-MS and at what concentration.
Line 128: . For clarity, define them at first use as “limit of detection (LOD)” and “limit of quantification (LOQ).”
Line 137: The approach to ANOVA post-hoc testing is unusual. You state that based on homogeneity of variance test results, either Duncan’s or Tukey’s test was chosen for post hoc analysis. The manuscript should justify the use of Duncan’s vs Tukey’s (what criterion was used to switch between them?) or, better, use a consistent method for all or a more appropriate method for unequal variances.
Line 137: The term “multiway ANOVA” is not standard; typically one would say “two-way ANOVA” for two factors (or multi-factor ANOVA).
Results
Line 150: Result 3.1. The sentence in the Results listing all the elements with significant regional differences is very long. For better readability, consider breaking it up or summarizing. For example: “Except for Cu, As, and Pb, all measured elements showed significant differences among the three origins (Li and several rare earth elements at p<0.05; the remaining 22 elements at p<0.01).
Line 156: Ensure that the description of significance aligns with the post-hoc comparisons. For instance, Zn is noted as highly significant among regions, and indeed the ANOVA p is <0.01. However, Table 2 shows that Bangdong and Ning’er have very close Zn values (39±1 vs 40±0, both marked “a”). This means the significant ANOVA was due to Jinggu’s lower Zn. It might be worth a brief note in the text to avoid confusion.
Line 167: Result 3.2. The phrasing “the content of As was statistically significant (p<0.05)” is not technically correct – content itself isn’t “significant.” It should be “the difference in As content was statistically significant.”
Line 206: Please add a space: “geographical origin discrimination.”
Line 178: Table 3 – Data on the exact region (production areas) of the tea analyzed for mineral composition at different processing stages are missing.
Line 172-177: Processing stage profiles – generalization. The text identifies specific stages at which certain elements “peaked” (e.g., Cs highest at kill-green, Nd at second turning, Sc/Co/Sm/Tm at fourth turning). This is a useful summary, but ensure that these trends were consistent across all three regions. Because you combined data for processing stages (implicitly averaging across regions), it’s possible some region’s pattern might differ.
Line 175: Element increases after fermentation. To strengthen this point, you could quantify it: for instance, K increased ~20% from fresh to final, Fe increased ~58% (86→136 mg/kg) etc., based on Table 3. Also, double-check if any listed element did not actually increase: e.g., Ba slightly decreased by final stage (from 27 to 23 mg/kg) and Cs peaked early then dropped by final.
In the PCA plot, four sample colors appear, but only three are defined in the legend. The legend should be completed.
Line 181: In PCA results, PC1 is said to “predominantly comprise” a large list of elements with 62.26% variance. This essentially indicates PC1 is an overall mineral content factor. PC2 and PC3 then highlight a few elements. It might be helpful to interpret these in plain language. For example: “PC1 had high loadings for nearly all elements, whereas PC2 and PC3 had high loadings on a subset of elements (e.g., Zn on PC2, and Cd, some rare earths on PC3). This suggests that while an overall multi-element gradient separates the teas (PC1, possibly correlating with overall soil mineral richness), there are secondary patterns driven by specific elements.”
Line 197: OPLS-DA model metrics. Consider mentioning how many components were used in the OPLS-DA model (e.g. one predictive and one orthogonal component?) and whether any samples were outliers. This detail can go in Methods or Results.
Line 197: VIP scores and key indicators. It might be useful to highlight that many of these VIP>1 elements are rare earth elements (Tb, Lu, Yb, Eu, etc.), which is consistent with prior studies suggesting rare earths are good geological tracers.
Line 205: The abbreviation should be defined at its first occurrence.
Discussion
Line 275-276: In the discussion of fermentation effects you mention elements like Cr, Ni, Be, Tl, Ag in citing Zhang et al.. Notably, your own analysis did not include Cr or Ni or those other elements (your ICP-MS list omitted them). Its important to address this so reviewers dont think it was an oversight.
Line 279-282: You note that according to multi-factor ANOVA, processing affected 27 elements (all except Tb), and the interaction affected 27 elements (all except Pb). This is intriguing because earlier, the one-way ANOVA (processing alone) found only 5 elements with significant differences among stages. The manuscript should address this apparent discrepancy. Why does the two-way ANOVA attribute significance to processing for almost all elements, whereas one-way ANOVA found only a few?
Line 288: The manuscript rightfully touts the 100% accuracy on the independent harvest-year validation (18 samples). To strengthen this, provide a bit more detail on those samples: How were the 18 validation samples distributed among the 3 regions?
.
Author Response
Dear Editor,
Thank you for giving us a chance to improve the quality of our review paper, those comments are all valuable and very helpful for revising and improving our paper. The whole manuscript was revised according to the reviewers’ and editor’s suggestions. Revised portions are marked in red on the paper.
Thank you very much for considering our manuscript. I'm looking forward to hearing from you soon.
Best regards,
Hongyan Liu Ph.D.
Research Center for Plants and Human Health, Institute of Urban Agriculture, Chinese Academy of Agricultural Sciences, Chengdu National Agricultural Science & Technology Center, Chengdu 610213, China.
E-mail: liuhongyan01@caas.cn
Reviewer: 3
Comments to the Author
The manuscript investigates the influence of geographic origin, fermentation processing, and their interaction on the multi-element composition of Pu-erh tea using ICP-MS, ANOVA, and multivariate statistical models to establish a reliable traceability approach validated across different harvest years. While the manuscript is generally interesting and holds sufficient interest for Foods, it has several limitations. I recommend a major revision of the text.
The title is somewhat long and can be more specific. The title uses the phrase “multi-elements,” which is grammatically odd. Consider rephrasing to “multi-element composition” or “multi-element profiles” for clarity and correctness or change title for example: “Effects of Region and Processing on Pu-erh Tea Elemental Profiles”.
- The titleis somewhat long and can be more specific. The title uses the phrase “multi-elements,” which is grammatically odd. Consider rephrasing to “multi-element composition” or “multi-element profiles” for clarity and correctness or change title for example: “Effects of Region and Processing on Pu-erh Tea Elemental Profiles”
Response: Thank you for your suggestion. We agree that “multi-elements” is not ideal and, as suggested, we have revised it to the more professional term “elemental profiles”. We have also added a new section (Section 3.3, Table S4) to specifically analyze the results of the two-way ANOVA. Since this new section highlights that the interaction between region and processing is a key finding of our study, we believe it is crucial to retain this term in the title to accurately reflect our work's contribution. The revised title is now: “Effects of region, processing and their interaction on the elemental profiles of Pu-erh tea”.
- Line 16: The wording “then processed with different processing methods” is repetitive and unclear.
Response: Thank you for your suggestion. This sentence has been modified. Please see line 15.
- Line 16-17:The sentence “significant differences were found in the content of 25 elements among different regions” is cumbersome. Consider: “significant regional differences were observed for 25 of the 28 elements.”
Response: Thank you for your suggestion. We have modified this sentence as recommended. Please see line 17.
- Line 19: For clarity, define abbreviation OPLS-DA at first use.
Response: Thank you for your suggestion. The abbreviation has been defined. Please see line 20
- Abstract vs. conclusion consistency (Lines 22-23 vs. Lines 305-307): The abstract mentions “100% accuracy through an independent validation set from different harvest years” (plural “years”), whereas the conclusion says “independent samples from a different year” (singular). This is slightly contradictory. If only one year of external samples was used (as it appears, 18 samples from the next year), change the abstract to “from a different harvest year” to match the actual procedure. Consistency here will avoid confusing readers about how many years of validation data were used.
Response: Thank you for your suggestion. The abstract has been modified for consistency. Please see line 24.
- The keywordsinclude terms already present in the title: Pu-erh tea
Response: Thank you for your suggestion. The keywords section has been modified.
- Line 30: Add “of tea” after “special variety” for clarity. Also, “found in Yunnan” could be “originating from Yunnan” to emphasize geographical origin.
Response: Thank you for your suggestion. This has been modified as requested. Please see line 31.
- Line 36: The participle “developing” does not agree with the subject. It should be “…making the tea taste smoother and develop a unique aged aroma.”
Response: Thank you for your suggestion. This has been modified. Please see line 37.
- Line 36-41: References are missing.
Response: Thank you for your suggestion. This part has been supplemented. Please see line 36-43.
- Line 59-62: The sentence is unclear; it should be rephrased and possibly split into two separate sentences.
Response: Thank you for your suggestion. This sentence has been modified. Please see line 61-64.
- Line 56-70:To strengthen the rationale of using multi-element profiles for origin authentication and to place the current study within a broader context of similar approaches applied to other plant-derived products, it is recommended to consider citing additional literature where ICP-MS and elemental fingerprinting were successfully used for the geographical discrimination of beverages (https://doi.org/10.3390/foods14020275).
Response: Thank you for your suggestion. This part has been added and modified. Please see line 73-75.
- Lines 30-79: The manuscript would benefit from an explicit statement of the knowledge gap and objective. Currently, many references are cited about tea origin authentication, but it’s not clearly stated what has not been done. Improve novelty.
Response: Thank you for your suggestion. This part has been added and modified. Please see line 76-91.
- Line 97: Sampling design clarity. It’s stated that 24 samples were collected per region (Table 1), and that leaves from each region were fermented separately. It’s unclear how those 24 samples were used in the fermentation experiment. Did each region’s 24 samples get combined into one pile, or were there multiple piles/replicates? Please clarify the experimental design: for example, if there were replicates of the fermentation per region or if those 24 samples correspond to, say, triplicate samples at each of 8 processing stages. This is important for understanding the statistical independence of the data.
Response: Thank you for your suggestion. This part has been added and modified. Please see line 122-128.
- Line 106: Awkward phrasing: “Placed in a dryer at 40℃until constant weight” should be rephrased for clarity. Typically one says “dried at 40 °C to constant weight.”
Response: Thank you for your suggestion. The phrasing has been modified. Please see line 130-131.
- Line 109: It should be “uniform particle size.”
Response: Thank you for your suggestion. This has been modified. Please see line 133.
- Line 127: Internal standard details. You mention that an internal standard was used. However, the specific internal standard element(s) are not identified anywhere. Please specify which internal standard(s) you used for ICP-MS and at what concentration.
Response: Thank you for your suggestion. This part has been added and modified. Please see line 155-156.
- Line 128: For clarity, define them at first use as “limit of detection (LOD)” and “limit of quantification (LOQ).”
Response: Thank you for your suggestion. We have modified this part. Please see line 160.
- Line 137: The approach to ANOVA post-hoc testing is unusual. You state that based on homogeneity of variance test results, either Duncan’s or Tukey’s test was chosen for post hoc analysis. The manuscript should justify the use of Duncan’s vs Tukey’s (what criterion was used to switch between them?) or, better, use a consistent method for all or a more appropriate method for unequal variances.
Response: Thank you for your suggestion. This part has been added and modified. Please see line 173-182.
- Line 137: The term “multiway ANOVA” is not standard; typically one would say “two-way ANOVA” for two factors (or multi-factor ANOVA).
Response: Thank you for your suggestion. We have modified this part. Please see line 170.
- Line 150:Result 3.1.The sentence in the Results listing all the elements with significant regional differences is very long. For better readability, consider breaking it up or summarizing. For example: “Except for Cu, As, and Pb, all measured elements showed significant differences among the three origins (Li and several rare earth elements at p<0.05; the remaining 22 elements at p<0.01).
Response: Thank you for your suggestion. We have modified this part. Please see line 192-194.
- Line 156: Ensure that the description of significance aligns with the post-hoc comparisons. For instance, Zn is noted as highly significant among regions, and indeed the ANOVA p is <0.01. However, Table 2 shows that Bangdong and Ning’er have very close Zn values (39±1 vs 40±0, both marked “a”). This means the significant ANOVA was due to Jinggu’s lower Zn. It might be worth a brief note in the text to avoid confusion.
Response: Thank you for your suggestion. We have modified this part. Please see line 195-197.
- Line 167:Result 3.2.The phrasing “the content of As was statistically significant (p<0.05)” is not technically correct – content itself isn’t “significant.” It should be “the difference in As content was statistically significant.”
Response: Thank you for your suggestion. We have modified this part. Please see line 210-211.
- Line 206: Please add a space:“geographical origin discrimination.”
Response: Thank you for your suggestion. We have modified this part. Please see line 232-233.
- Line 178: Table 3 – Data on the exact region (production areas) of the tea analyzed for mineral composition at different processing stages are missing.
Response: Thank you for your suggestion. This part of the data has been added to TableS3.
- Line 172-177: Processing stage profiles – generalization. The text identifies specific stages at which certain elements “peaked” (e.g., Cs highest at kill-green, Nd at second turning, Sc/Co/Sm/Tm at fourth turning). This is a useful summary, but ensure that these trends were consistent across all three regions. Because you combined data for processing stages (implicitly averaging across regions), it’s possible some region’s pattern might differ.
Response: Thank you for your suggestion. This part of the data has been added to Table S3, and the relevant analysis has been added to Section 3.2. Please see lines 222-233.
- Line 175: Element increases after fermentation. To strengthen this point, you could quantify it: for instance, K increased ~20% from fresh to final, Fe increased ~58% (86→136 mg/kg) etc., based on Table 3. Also, double-check if any listed element did not actually increase: e.g., Ba slightly decreased by final stage (from 27 to 23 mg/kg) and Cs peaked early then dropped by final.
Response: Thank you for your suggestion. This part has been added and modified. Please see line 213-221.
- In the PCA plot, four sample colors appear, but only three are defined in the legend. The legend should be completed.
Response: Thank you for your suggestion. The PCA plot has been revised and beautified. Please see line 261.
- Line 181: In PCA results, PC1 is said to “predominantly comprise” a large list of elements with 62.26% variance. This essentially indicates PC1 is an overall mineral content factor. PC2 and PC3 then highlight a few elements. It might be helpful to interpret these in plain language. For example: “PC1 had high loadings for nearly all elements, whereas PC2 and PC3 had high loadings on a subset of elements (e.g., Zn on PC2, and Cd, some rare earths on PC3). This suggests that while an overall multi-element gradient separates the teas (PC1, possibly correlating with overall soil mineral richness), there are secondary patterns driven by specific elements.”
Response: Thank you for your suggestion. This part has been added and modified. Please see line 250-256.
- Line 197: OPLS-DA model metrics. Consider mentioning how many components were used in the OPLS-DA model (e.g. one predictive and one orthogonal component?) and whether any samples were outliers. This detail can go in Methods or Results.
Response: Thank you for your suggestion. We have re-verified this part of the data, corrected the content in the manuscript, and revised the relevant section. Please see lines 272-280.
- Line 197: VIP scores and key indicators. It might be useful to highlight that many of these VIP>1 elements are rare earth elements (Tb, Lu, Yb, Eu, etc.), which is consistent with prior studies suggesting rare earths are good geological tracers.
Response: Thank you for your suggestion. This part has been added and modified. Please see line 278-280.
- Line 205: The abbreviation should be defined at its first occurrence.
Response: Thank you for your suggestion. This part has been added and modified. Please see line 272.
- Line 275-276: In the discussion of fermentation effects you mention elements like Cr, Ni, Be, Tl, Ag in citing Zhang et al.. Notably, your own analysis did not include Cr or Ni or those other elements (your ICP-MS list omitted them). Its important to address this so reviewers dont think it was an oversight.
Response: Thank you for your suggestion. To improve the clarity and focus of our discussion, we have revised this sentence (please see Lines 351-352 in the revised manuscript), only the elements directly relevant to our own findings (i.e., Ba, Li, and Sr) were cited. We believe this modification makes the comparison more direct and the text more concise.
- Line 279-282: You note that according to multi-factor ANOVA, processing affected 27 elements (all except Tb), and the interaction affected 27 elements (all except Pb). This is intriguing because earlier, the one-way ANOVA (processing alone) found only 5 elements with significant differences among stages. The manuscript should address this apparent discrepancy. Why does the two-way ANOVA attribute significance to processing for almost all elements, whereas one-way ANOVA found only a few?
Response: Thank you for your suggestion. We have included the full results of our two-way ANOVA in a new section, “3.3 Effects of region, processing and their interaction on multi-elements,” supported by a detailed Table S4. Furthermore, we have added a detailed explanation for this statistical phenomenon in the Discussion section (Lines 369-386). The key reason is the difference in how the models handle variance. In the one-way ANOVA, all samples from the three regions were analyzed together. The very large variance introduced by the 'region' factor likely masked the more subtle effect of 'processing'. In contrast, the two-way ANOVA mathematically partitions the variance, distinguishing the dominant effect of region from the more subtle effect of processing. This makes it far more sensitive and much larger number of elements were found to be influenced by processing.
- Line 288: The manuscript rightfully touts the 100% accuracy on the independent harvest-year validation (18 samples). To strengthen this, provide a bit more detail on those samples: How were the 18 validation samples distributed among the 3 regions?
Response: Thank you for your suggestion. We have provided an explanation for this section in the manuscript. Please see lines 391-392.
Round 2
Reviewer 3 Report
Comments and Suggestions for Authors
The authors have addressed all of my comments, and I am satisfied with their responses.
However, I kindly ask the authors to carefully review all references, as some of them contain incorrect author listings. For instance:
- In reference 8, the authors are incorrectly listed as Ongiéras, F.; George, C.; Lizzani-Cuvelier, L.; Loiseau, A.-M.; Fernandez, X., while Lydia Gautier and Mickael Le Bechec are missing.
- In reference 9, the authors are incorrectly listed as Chen, D.-L.; Liu, S.-Y.; Xu, Y.-Q., and Li-Yong Luo and Liang Zeng have been omitted.
- In reference 25, the authors are incorrectly listed as Jović, M.; Eger, A.M.; Gąstoł, M.; Jurczak, M.; Momčilović, M., while Onjia, A. has been omitted.
- In reference 26, the listed authors—Giraudo, A.; Iezzi, A.; Ivaldi, C.; Panero, L.; Rolle, L.; Zago, M.; Gerbi, V.—are incorrect, and Christos Tsolakis and Aldo Ciambotti have been omitted.
Please check all the references thoroughly, as only a few examples have been provided here.
Author Response
Dear Editor,
Thank you for considering our revised manuscript. We appreciate the valuable feedback from you and the reviewers, which has significantly improved the quality of our paper. The references section was revised according to the reviewers’ and editor’s suggestions. Revised portions are marked in red on the paper.
Thank you very much for considering our manuscript. I'm looking forward to hearing from you soon.
Best regards,
Hongyan Liu Ph.D.
Research Center for Plants and Human Health, Institute of Urban Agriculture, Chinese Academy of Agricultural Sciences, Chengdu National Agricultural Science & Technology Center, Chengdu 610213, China.
E-mail: liuhongyan01@caas.cn
Reviewer: 3
Comments to the Author
The authors have addressed all of my comments, and I am satisfied with their responses.
However, I kindly ask the authors to carefully review all references, as some of them contain incorrect author listings. For instance:
In reference 8, the authors are incorrectly listed as Ongiéras, F.; George, C.; Lizzani-Cuvelier, L.; Loiseau, A.-M.; Fernandez, X., while Lydia Gautier and Mickael Le Bechec are missing.
In reference 9, the authors are incorrectly listed as Chen, D.-L.; Liu, S.-Y.; Xu, Y.-Q., and Li-Yong Luo and Liang Zeng have been omitted.
In reference 25, the authors are incorrectly listed as Jović, M.; Eger, A.M.; Gąstoł, M.; Jurczak, M.; Momčilović, M., while Onjia, A. has been omitted.
In reference 26, the listed authors—Giraudo, A.; Iezzi, A.; Ivaldi, C.; Panero, L.; Rolle, L.; Zago, M.; Gerbi, V.—are incorrect, and Christos Tsolakis and Aldo Ciambotti have been omitted.
Please check all the references thoroughly, as only a few examples have been provided here.
Response: Thank you for your valuable feedback. We have carefully reviewed your comments regarding the references, and also meticulously checked and corrected all author listings in the reference section. As requested, all revised parts have been marked in red in the manuscript.